# Minimal Residual Disease in Colorectal Cancer: Are We Finding the Needle in a Haystack?

**DOI:** 10.3390/cells12071068

**Published:** 2023-04-01

**Authors:** Alexandre A. Jácome, Benny Johnson

**Affiliations:** 1Department of Gastrointestinal Medical Oncology, Oncoclinicas, Belo Horizonte 30360-680, Brazil; alexandre.jacome@medicos.oncoclinicas.com; 2Department of Gastrointestinal Medical Oncology, Division of Cancer Medicine, The University of Texas MD Anderson Cancer Center, Houston, TX 77030, USA

**Keywords:** colorectal neoplasms, circulating tumor DNA, neoplasm, residual, chemotherapy, adjuvant, high-throughput nucleotide sequencing

## Abstract

Despite significant advances in the surgical and systemic therapy of colorectal cancer (CRC) in recent decades, recurrence rates remain high. Apart from microsatellite instability status, the decision to offer adjuvant chemotherapy to patients with CRC is solely based on clinicopathologic factors, which offer an inaccurate risk stratification of patients who derive benefit from adjuvant therapy. Owing to the recent improvements of molecular techniques, it has been possible to detect small allelic fractions of circulating tumor DNA (ctDNA), and therefore, to identify patients with minimal residual disease (MRD) after curative-intent therapies. The incorporation of ctDNA identifying MRD in clinical practice may dramatically change the standard of care of CRC, refining the selection of patients who are candidates for escalation and de-escalation of adjuvant chemotherapy, and even for organ-preservation strategies in rectal cancer. In the present review, we describe the current standard of care and the DNA sequencing methodologies and assays, present the data from completed clinical studies and list ongoing potential landmark clinical trials whose results are eagerly awaited, as well as the impact and perspectives for the near future. The discussed data bring optimism for the future of oncologic care through the hope of refined utilization of adjuvant therapies with higher efficacy and safety for patients with both localized and advanced CRC.

## 1. Introduction

Colorectal cancer (CRC) incidence and mortality rates have decreased in recent decades in the US, but it remains as the third leading cause of cancer-related death [1]. The 5-year overall survival (OS) of the patients is stage dependent: 91%, 72%, and 15%, in localized, regional, and metastatic disease, respectively [1,2].

Despite significant recent improvements in the systemic therapy of metastatic CRC (mCRC), the recurrence rates after surgical treatment of localized disease remain high. It is estimated that approximately 20% of stage II and 35% of stage III patients will develop disease recurrence within 5 years of follow up, even with the use of optimal adjuvant chemotherapy [3,4].

Modest advances have been made in the adjuvant therapy of colon cancer in recent decades. Oxaliplatin-based adjuvant chemotherapy is offered to stage III and some high-risk stage II patients, while fluoropyrimidine alone is the adjuvant therapy of choice for stage II patients who are deemed as high risk [5]. The selection of patients who presumably derive larger benefit from adjuvant chemotherapy is based on the presence of traditional prognostic clinicopathologic factors, which offer an inaccurate risk stratification, and therefore, are associated with both overtreatment and undertreatment.

The clinicopathologic adverse risk factors, such as T4 disease, fewer than 12 examined lymph nodes, angiolymphatic or perineural invasion, poorly differentiated tumors, high-score tumor budding, and tumors presenting with obstruction and perforation, are proxy of micrometastatic disease [5,6]. The detection of circulating tumor cells or circulating tumor DNA (ctDNA) in very small plasmatic concentrations has never been possible until the development of modern techniques, which may identify patients with postoperative minimal residual disease (MRD) who will ultimately recur without additional therapy, and therefore, better select patients for adjuvant chemotherapy.

The advent of this technology may dramatically change the landscape of adjuvant therapy for colon cancer. Prospective and retrospective studies have demonstrated high specificity of the test and a recent randomized clinical trial also demonstrated important therapeutic implications of the detection of postoperative MRD.

In the present review, we aim to describe the unmet clinical need of personalization and precision with adjuvant therapy for colon cancer, the strength and limitations of the current state of the detection of MRD, the completed and ongoing studies evaluating postoperative ctDNA, and perspectives regarding the clinical applicability of the detection of MRD in colon cancer over the next few years.

## 2. Standard of Care of Adjuvant Therapy of Colon Cancer

Approximately 20% of stage II and 35% of stage III patients with colon cancer will present with disease recurrence within 5-year follow up despite the use of optimal adjuvant chemotherapy [3,4]. The population of patients with stage II disease comprise a heterogeneous group with different risks of recurrence. The selection of stage II patients who presumably derive larger benefit from adjuvant chemotherapy is based on the presence of prognostic clinicopathologic factors, which offer an inaccurate risk stratification in low- and high-risk stage II colon cancer (Figure 1) [6]. There is no consensus among NCCN, ASCO, and ESMO regarding the clinicopathologic factors that should be considered in risk stratification [6,7,8]. It is generally accepted that stage II patients with T4 tumors, fewer than 12 examined lymph nodes, angiolymphatic or perineural invasion, poorly differentiated tumors, bowel obstruction and perforation at presentation, and high-score tumor budding should be considered for six months of fluoropyrimidine-based adjuvant chemotherapy, either 5-FU plus folinic acid or capecitabine. The routine use of adjuvant chemotherapy in all patients with stage II colon cancer is not recommended, since the supposed absolute improvement of approximately 2% in OS does not outweigh the risks imposed by the acute and late fluoropyrimidine-related adverse events [7]. Nevertheless, it is supposed that some stage II patients, mainly those with T4N0 disease or with fewer than 12 examined lymph nodes, present higher risk of recurrence than some stage III patients (T1-T2N1a) [9]. In these selected stage II patients, the use of oxaliplatin may be considered in the adjuvant therapy [6].

Approximately 15% of stage II patients present tumors that harbor deficient mismatch repair (dMMR)/high-frequency microsatellite instability (MSI-H) [10]. These patients have a lower risk of recurrence and do not derive benefit from adjuvant chemotherapy composed of fluoropyrimidine alone [10,11]. Those patients with high-risk features, such as T4 disease and less than 12 examined lymph nodes, doublet chemotherapy may be considered for at least three months [6].

Oxaliplatin-based adjuvant chemotherapy is routinely recommended in stage III colon cancer [4]. Since 2004, 6-month oxaliplatin-based adjuvant chemotherapy has been the standard of care of node-positive colon cancer, but peripheral neuropathy, which may persist in approximately 15% of the patients for over four years, is the most limiting oxaliplatin-related adverse event [12]. In order to offer a safer alternative, IDEA collaboration demonstrated that stage III patients classified as low-risk (T3N1) present an acceptable minimal loss of efficacy if exposed to 3-month CAPOX compared to 6-month FOLFOX or CAPOX, and therefore, may be considered to a shorter duration of adjuvant chemotherapy [13]. Patients classified as high-risk stage III (T4 and/or N2) would have a higher loss of efficacy with 3-month oxaliplatin-based regimen, mainly 3-month FOLFOX, and should be considered for 6-month adjuvant chemotherapy. ACCENT database also suggests that the magnitude of benefit of the addition of adjuvant oxaliplatin decreases in older patients [14]. Based on an exploratory analysis of subgroups of randomized clinical trials, patients older than 70 years old do not derive benefit from the addition of oxaliplatin in the adjuvant chemotherapy, mainly if greater than 80 years old. Hence, the decision to offer adjuvant oxaliplatin to elderly patients should be made as a case-by-case basis.

Stage III patients derive similar benefit from oxaliplatin-based adjuvant chemotherapy regardless of MSI status [15,16]. Other predictive biomarkers which have therapeutic implications in metastatic disease, such as RAS, BRAF, and HER2, do not currently influence the decision regarding adjuvant therapy. Despite having demonstrated meaningful efficacy in MSI-H mCRC, immune checkpoint inhibitors (ICI) have not yet demonstrated efficacy in adjuvant therapy. Results of ongoing ATOMIC clinical trial (NCT02912559) evaluating ICI in combination with chemotherapy for MSI-H stage III colon cancer are eagerly awaited in the forthcoming years.

## 3. DNA Sequencing Methodologies

When normal cells undergo apoptosis, their DNA is shed into the bloodstream in small fragments composed of approximately 180–200 base pairs [17]. The pool of circulating DNA, which is called cell-free DNA (cfDNA), is usually steady, but it can increase in pathologic conditions, such as cellular injuries or necrosis [18]. Cancer cells are characterized by high proliferation index and fast turnover. The higher the volume of the disease in a patient, the higher the amount of tumor DNA shed into the bloodstream [19]. The DNA from cancer cells detected in the bloodstream is called circulating tumor DNA (ctDNA), whose percentage in the pool of cfDNA (ctDNA/cfDNA) is denominated allelic fraction, which may vary according to the burden of the disease, from 0.1% to >90% [18].

ctDNA is discriminated from normal DNA by sequencing techniques, which detect the presence of mutations, typically found in ctDNA. Standard DNA sequencing techniques, such as Sanger sequencing and pyrosequencing, may detect mutations in small fragments of ctDNA, but only if high allelic fractions are present, such as >10% [18,20]. The detection of smaller allelic fractions has been challenging, and it has been possible by the development of modern techniques, such as digital polymerase chain reaction (PCR) and beads, emulsion, amplification, and magnetics (BEAMing) (Figure 2) [18,20].

The detection of ctDNA in postoperative setting, searching for MRD, is especially challenging, since it may be truly absent or be present in very small allelic fractions. Tumor-informed assays have been developed as a tool to increase the sensitivity of the test, mainly to distinguish from potential false positives, such as clonal hematopoiesis of indeterminate potential (CHIP) [21,22]. Whole-exome sequencing (WES) is performed on matched formalin-fixed paraffin embedded (FFPE) tumor DNA and germline DNA from leucocytes. Somatic structural variants (SSV) and somatic point mutations (SPM) are detected in tumor DNA and they will form the library [23,24]. The genes selected may vary according to the tumor-informed assay. One prospective study analyzed 15 genes (*SMAD4*, *TP53*, *AKT1*, *APC*, *BRAF*, *CTNNB1*, *ERBB3*, *FBXW7*, *HRAS*, *KRAS*, *NRAS*, *PIK3CA*, *PPP2R1A*, *RNF43*, and *POLE*). Digital droplet PCR (ddPCR) assays targeting those SSV and SPM are designed using primers, which will be used in serial plasma samples. Each analysis includes a matched tumor DNA (positive control), germline DNA (negative control), and a non-template control [23,24]. Tumor-informed assays have provided specificity of 100%, but with unsatisfactory sensitivity, as we describe in the next sections.

Availability of tumor tissue to perform a tumor-informed assay may limit the test. Neoadjuvant therapies have been more frequently used in recent years, and surgical specimens may offer an inadequate tumor tissue for molecular tests in cases of pathologic responses [25]. Tumor-informed ctDNA assays for the detection of MRD are in development and, currently, there are few commercially available options. A plasma-only ctDNA assay integrating genomic and epigenomic cancer signatures to enable tumor-uninformed MRD detection might overcome the unavailability of tumor tissue and it has been evaluated in recent studies [25,26], which will be described in the next section.

Regardless of the technique, ctDNA assays for the detection of MRD will require high specificity to escalate adjuvant therapy in patients who would be categorized as low-risk according to clinicopathologic factors and would ultimately recur without additional therapy, and high sensitivity to allow safe de-escalation of therapy in those patients who are traditionally treated with intensive chemotherapy and avoidable adverse events.

## 4. Completed Studies

Owing to the high allelic fractions of ctDNA in metastatic disease, the completion of clinical studies in patients with advanced disease is less complex compared to studies involving detection of MRD. Therefore, ctDNA has been progressively used in the management of mCRC in recent years, mainly for RAS/BRAF molecular profiling and for anti-EGFR rechallenge strategies. The conceptualization of studies addressing the clinical applicability of ctDNA as MRD in colon cancer has only been possible after the development of molecular assays with higher sensitivity and specificity, such as the tumor-informed and plasma-only assays (Table 1).

A prospective study involving 130 patients with stage I to III CRC evaluated the prognostic role of postoperative ctDNA as MRD [24]. Blood samples were longitudinally collected before surgery (up to 14 days preoperatively), at postoperative day 30, and every third month until death, patient withdrawal from the study, or month 36, whichever came first. ctDNA assay was based on tumor whole-exome sequencing, generating 16 high-ranked patient-specific somatic single-nucleotide variants and short indels for every patient, and multiplex PCR primer pairs were generated and applied in the extracted plasma. Preoperatively, 108 out of 122 patients (88.5%) presented as ctDNA positive. At postoperative day 30, 10 out of 94 patients (10.6%) were ctDNA positive, and 7 of them presented recurrence and shorter DFS compared to ctDNA-negative counterparts (HR: 7.2, 95% CI 2.7–19.0, *p* < 0.001). The relapse rate for ctDNA-negative patients was 12% (10 of 84 patients). All of the 10 ctDNA-positive patients at postoperative day 30 received adjuvant chemotherapy, and 3 of them had ctDNA cleared and were disease free at the end of the follow up, while 7 relapsed. The positivity for ctDNA at the conclusion of the adjuvant chemotherapy was a strong prognostic factor. Of the 58 patients with available blood samples after adjuvant chemotherapy, 7 were ctDNA positive and all of them presented relapse (100%). The relapse rate in the remaining 51 patients was 14% (HR: 17.5, 95% CI 5.4–56.5, *p* < 0.001). In the surveillance period, 75 patients had availability of blood samples. Of the 15 ctDNA-positive patients, 14 presented recurrence (93.3%), compared to only 2 of the remaining 60 ctDNA-negative patients (HR: 43.5, 95% CI 9.8–193.5, *p* < 0.001). Interestingly, this study also showed that positivity for ctDNA allowed for an earlier diagnosis of relapse compared to the standard of care computed tomography, with a mean lead time of 8.7 months (range 0.8–16.5 months).

CIRCULATE-Japan project is a large platform enrolling patients with stage II to IV resectable CRC aiming to evaluate the clinical utility of ctDNA MRD analysis. It is composed of one observational study (GALAXY) and two randomized phase III trials (VEGA and ALTAIR studies). Data from GALAXY were recently presented [30]. A population of 1040 patients were included: 9% stage I, 30% stage II, 38% stage III, and 22% stage IV. Blood samples were collected before surgery and 4, 12, 24, 36, 48, 72, and 96 weeks after surgery. Patients were categorized in two groups: post-operative 4 weeks ctDNA negative (N = 852) and ctDNA positive (N = 188). In a median follow-up time of 11.4 months, 12-month DFS was 92.7% versus 47.5%, respectively (HR: 10.9, 95% CI 7.8–15.4, *p* < 0.001), with a sensitivity of 63.6%. When stage II–III population was analyzed, 12-month DFS was 95.2% versus 55.5% (HR: 13.3, 95% CI 8.0–22.2, *p* < 0.001), with a sensitivity of 67.6%. In multivariate analysis, positivity for ctDNA was the strongest prognostic factor (HR: 15.3, 95% CI 8.6–27.2, *p* < 0.001). From the overall population of 1040 patients, a cohort of 838 had their DFS analyzed by ctDNA dynamics from post-operative 4 weeks to 12 weeks. The 6-month DFS of patients who had clearance of ctDNA (positive at 4w > negative at 12w) was comparable to the ctDNA-negative counterparts (negative > negative): 98% versus 100%, respectively. The effect of adjuvant chemotherapy on the ctDNA clearance rate was evaluated in the population of patients who presented as ctDNA positive after surgical treatment. From 183 patients, 96 received adjuvant chemotherapy and 87 did not receive. The clearance rate was 68% versus 10% (HR: 9.3, 95% CI 4.6–18.9, *p* < 0.001). In multivariate analysis, adjuvant chemotherapy was the strongest prognostic factor for recurrence in the ctDNA-positive population (HR: 5.6, 95% CI 3.2–9.7, *p* < 0.001). Adjuvant chemotherapy had no effect on the recurrence rate in the 531 patients who presented as ctDNA negative after surgical treatment.

DYNAMIC was the first randomized clinical trial addressing ctDNA-guided approach in the therapeutic management of stage II colon cancer [34]. Using a tumor-informed assay for the detection of MRD, the study collected plasma specimens at postoperative weeks 4 and 7 before randomization, which assigned 455 patients in a 2:1 ratio to two approaches: ctDNA guided versus standard. In the ctDNA-guided approach, patients who were ctDNA positive received adjuvant chemotherapy (fluoropyrimidine alone or oxaliplatin-based chemotherapy) and those negative did not receive adjuvant chemotherapy. In the standard approach arm, patients received adjuvant chemotherapy according to the standard clinicopathologic criteria. The median follow up was 37 months. In the analysis of recurrence-free survival at 2 years, the primary endpoint of the study, ctDNA-guided approach was non-inferior to the standard approach (93.5% versus 92.4%, respectively). In addition, a lower percentage of the patients in ctDNA arm received adjuvant chemotherapy compared to the standard arm: 15% versus 28%, respectively (RR: 1.82, 95% CI 1.25–2.65). These findings are provocative in that clinical management guided by ctDNA as a predictive biomarker for adjuvant chemotherapy without compromising risk of recurrence for patients provides an alternative path from overtreatment and needless long-term toxicity. Nevertheless, the results of DYNAMIC trial should be analyzed with caution. First, patients in the ctDNA arm were more intensively treated than patients in the standard arm: the percentage of patients treated with oxaliplatin-based adjuvant chemotherapy was 62% versus 10%, respectively. Second, ctDNA-negative patients who were pT4 presented similar recurrence risk compared to ctDNA-positive patients who received adjuvant chemotherapy (HR: 3.04, 95% CI 1.26–7.34 versus HR: 3.69, 95% CI 1.39–9.87, respectively). Third, likewise, ctDNA-negative patients who would have been categorized as high risk according to clinicopathologic criteria also showed similar recurrence risk compared to ctDNA-positive patients who received adjuvant chemotherapy (HR: 2.60, 95% CI 1.01–6.71 versus HR: 2.62, 95% CI 1.11–6.20, respectively). The DYNAMIC trial has provided a greater understanding of the evolving role of ctDNA to define MRD in the therapeutic management of stage II colon cancer and represents a significant leap forward in the incorporation of ctDNA in clinical practice. The prospect of having a biomarker for clinical ‘go’ or ‘no go’ decisions regarding use of adjuvant chemotherapy in stage II colon cancer is definitely a step towards true precision for patients. However, it also highlighted that even while adopting a tumor-informed assay, the sensitivity of the test was not high enough to detect a percentage of the ctDNA-negative patients who developed subsequent disease recurrence (false-negative patients). Therefore, the trial suggests that clinicopathologic adverse risk factors may continue to serve as an adjunct to ctDNA for refined selection of patients for adjuvant chemotherapy.

Plasma-only assays have also been evaluated in recent studies [25,26]. From an initial sample of 103 patients with CRC, 84 (from stage I to IV) had evaluable plasma drawn after completion of definitive therapy (surgery or completion of adjuvant therapy) [25]. All 15 patients with detectable ctDNA presented recurrence (100% of positive predictive value). Of 49 patients without detectable ctDNA, 12 recurred, leading to a sensitivity of 55% and a specificity of 100%. Integrating epigenomic signatures increased sensitivity by 25% to 36% versus genomic alterations alone, reaching up to 91% of sensitivity. This assay has been currently used in the NRG cooperative group clinical trial COBRA (NCT04068103), which aims to evaluate the role of ctDNA in guiding treatment decisions in stage IIA colon cancer.

The use of techniques that detect epigenomic signatures assumes that hypermethylation of promoter regions of some tumor suppressor genes is a common finding in CRC. Hypermethylation of WNT-inhibitor-factor-1 (WIF1) and neuropeptide Y (NPY) has been validated as a marker of both advanced and localized colon cancer [35,36]. A post hoc analysis of the IDEA France clinical trial analyzed ctDNA of 1017 stage III colon cancer patients who had been randomized to 3-month versus 6-month oxaliplatin-based adjuvant chemotherapy, testing for WIF1 and NPY by droplet digital PCR [26]. There were 877 (86%) ctDNA-negative and 140 (14%) ctDNA-positive patients in samples collected before the beginning of adjuvant chemotherapy. ctDNA-positive patients presented a 3-year DFS of 66.39% versus 76.71% for ctDNA-negative counterparts (*p* = 0.015). In addition, ctDNA was an independent prognostic factor for DFS (HR:1.55, 95% CI 1.13–2.12, *p* = 0.006) and OS (HR:1.65, 95% CI 1.12–2.43, *p* = 0.011). These two studies evaluating plasma-only ctDNA assays show the potential clinical applicability of tumor-uninformed tests based on the detection of epigenomic signatures.

## 5. MRD in Rectal Cancer

The therapeutic management of rectal cancer located above the peritoneal reflection is similar to that applied in colon cancer. Nevertheless, neoadjuvant therapies are routinely recommended in the management of locally advanced rectal cancer (LARC) located at the low- or mid-rectum. Given the high morbidity associated with the surgical treatment of rectal cancer and the favorable recent data of the potential role of total neoadjuvant therapy in organ-preservation strategy, non-operative management (NOM) has been progressively considered in the treatment of rectal cancer. The selection of patients for NOM is challenging, since the prediction of pathologic complete response (pCR) is inaccurate by using non-invasive methods, such as digital rectal examination, magnetic resonance imaging (MRI), and colonoscopy.

Correlative studies evaluating patients with LARC have been concordant in showing association between ctDNA levels and clinical response to either chemoradiotherapy (CRT) or surgery [37,38,39,40,41,42]. Additionally, four studies demonstrated correlation between ctDNA levels and tumor regression grade (TRG) score in MRI after CRT [38,39,42,43], and another one showed association between involvement of circumferential resection margin and detection of ctDNA by a methylation-based assay [44]. However, studies have showed discordant results in demonstrating the value of ctDNA for predicting pCR [40,41,45,46]. Prospective study with 159 patients with LARC using a tumor-informed assay detected ctDNA in 77%, 8%, and 12% of pre-CRT, post-CRT, and postoperative plasma samples [45]. The prognostic value of ctDNA in predicting recurrence-free survival (RFS) was consistently demonstrated if detected both after CRT (HR: 6.6, *p* < 0.001) and after surgery (HR: 13.0, *p* < 0.001). The estimated 3-year RFS was 87% for postoperative ctDNA negative versus 33% for postoperative ctDNA positive. Nevertheless, there was no correlation between post-CRT ctDNA status and pCR. From the 112 patients who were pre-CRT ctDNA positive, 21 (18%) presented pCR. ctDNA turned negative in 20/21 patients (95%), but it also did in 80/91 (88%) patients who did not reach pCR [45]. On the other hand, prospective study with 61 patients with LARC showed a correlation between post-CRT/preoperative ctDNA status and pathological downstaging. In addition, preoperative ctDNA positivity was associated with node-positive disease in the surgical specimen [46]. ctDNA dynamics after neoadjuvant therapy, mainly after radiation therapy, needs to be elucidated before adopting the detection of MRD as a guide for therapeutic decision after CRT in the treatment of rectal cancer. Likewise, the persistence of ctDNA positivity after trimodality therapy in rectal cancer has been demonstrated to be a poor prognostic factor, but the potential impact of adjuvant chemotherapy based on post-treatment ctDNA status should be evaluated in randomized clinical trials.

## 6. MRD in Metastatic Colorectal Cancer

Surgical resection of CRC liver metastases (CRLM) confined to the organ is a potentially curative therapy and offers superior long-term disease control compared to non-surgical therapies [47,48,49]. Perioperative chemotherapy prolongs disease-free survival, but the benefit on overall survival (OS) has never been demonstrated [50,51]. Therefore, there is no consensus on which mCRC patients should undergo postoperative systemic therapy. The identification of MRD after liver resection might help in the appropriate selection of patients for systemic therapy.

Several prospective studies have confirmed the strong prognostic value of postoperative MRD after resection of CRLM [31,52,53,54,55,56,57]. In a prospective study with 112 mCRC patients who had undergone liver resection with curative intent, ctDNA positivity by a tumor-informed assay was the most significant prognostic factor for disease-free survival (HR: 5.7, 95% CI 3.3–10.0) [52]. MRD was detected in 61 (54%) patients, of which 59 (97%) presented recurrence at the time of data cutoff. At the time of analyses, 96% (49/51) of patients were alive in the MRD-negative arm compared with 52% (32/61) in the MRD-positive arm [52]. Likewise, another prospective study with 48 mCRC patients who underwent preoperative chemotherapy followed by resection of CRLM collected blood samples before and after hepatectomy [57]. ctDNA by a plasma-only assay was detected before and after surgery (ctDNA+/+) in 14 (29%) patients, before but not after surgery (ctDNA+/−) in 19 (40%), and not at all (ctDNA−/−) in 11 (23%). ctDNA+/+ was associated with worse RFS (median 6.0 months), compared to ctDNA+/− (median not reached), and ctDNA−/− (median 33.0 months; *p* = 0.001). In multivariate analyses, ctDNA+/− and ctDNA−/− were independently associated with improved RFS compared to ctDNA+/+ (ctDNA+/−: HR 0.21, 95% CI 0.08–0.53; ctDNA−/−: HR 0.21, 95% CI 0.08–0.56) [57]. ctDNA status has also been demonstrated to be a prognostic factor after local therapies of CRLM, such as stereotactic radiotherapy and radiofrequency ablation [58]. Another prospective study with 105 patients with CRLM who underwent curative-intent hepatectomy and tested for ctDNA within 180 days postoperatively showed that a postoperative ctDNA-positive result was associated with multiple CRLM and with co-mutation RAS/TP53 [56].

Ongoing clinical trials (Table 2) will elucidate if these high-risk ctDNA-positive patients derive benefit from interventional strategies. Based on the rationale that patients with micrometastatic microsatellite stable CRC presents increased TGF-β signaling and exclusion of anti-tumor cytotoxic T cells in the tumor microenvironment, the use of bintrafusp alfa, a bifunctional fusion protein composed of the extracellular domain of the TGF-βRII receptor and anti-PD-L1 antibody, was evaluated in patients who had undergone resection of CRLM and presented postoperative ctDNA positivity. However, the use of bifunctional fusion protein did not improve the ctDNA clearance rate [27].

## 7. Ongoing Clinical Trials

The completed clinical studies aforementioned have demonstrated the powerful prognostic value of MRD after curative-intent therapies in the management of colorectal cancer. Patients who present ctDNA positivity after surgical resection, and, mainly after the completion of adjuvant chemotherapy, portend poor RFS. The DYNAMIC and the ongoing clinical trials address therapeutic interventions in the group of patients who show MRD after curative-intent therapies and have ctDNA clearance rate and survival outcomes as the main objectives (Table 2).

## 8. Conclusions

Despite recent advances in surgical and systemic therapies of colorectal cancer, the recurrence rates of localized disease remain concerningly high. From a molecular standpoint, CRC is a markedly heterogeneous disease, but clinical decisions regarding adjuvant therapy are solely based on clinicopathologic factors, apart from MSI status, which is paramount in the decision to offer adjuvant chemotherapy in stage II patients. Clinicopathologic factors are only proxy of micrometastatic disease so that they do not offer an accurate risk stratification of patients who would derive benefit from adjuvant therapy. The possibility to detect ctDNA after curative-intent therapies is a proof of MRD, and the clinical studies have demonstrated that, concordant with biological plausibility, these ctDNA-positive patients will ultimately recur without additional therapies.

Nevertheless, these clinical studies have demonstrated that some clinicopathologic factors, such as T4 tumors, remain clinically significant and prognostic, even in ctDNA-negative patients. Based on the clinical data so far, even with the improvement of molecular techniques, it is likely that the sensitivity of the test will not reach the desired high levels in the therapeutic window of 4–8 weeks used for the decision to offer adjuvant chemotherapy. Studies have shown that serial monitoring in the surveillance period significantly increases the sensitivity of the test, but the clinical value of ctDNA positivity several months or years after the completion of adjuvant chemotherapy is not clear. The comparative benefit of starting systemic therapy with palliative intent guided by ctDNA dynamics or radiologic findings have not yet been addressed. Hence, it is possible that the combination of molecular (ctDNA) and clinical data might be a reasonable strategy to offer a more personalized therapy for patients with localized colorectal cancer (Figure 3).

Many questions still demand answers before routinely adopting tests for the detection of MRD in the therapeutic management of CRC: (1) Should we escalate systemic therapy to oxaliplatin-based adjuvant chemotherapy in ctDNA-positive low-risk stage II colon cancer? (2) If so, how long? For 3 months or 6 months? (3) Should we de-escalate adjuvant chemotherapy in ctDNA-negative high-risk stage II and stage III patients? (4) How do we manage patients who persistently show ctDNA positivity after the completion of adjuvant chemotherapy? (5) How to manage patients who turn ctDNA positive in the surveillance period? Currently, based on the high specificity of the tumor-informed assays, while it is plausible to offer at least 3-month oxaliplatin-based adjuvant chemotherapy for a low-risk stage II patient who is ctDNA positive after curative-intent surgical resection considering the lack of data in this space, clinical trial enrollment to further understand the management of MRD in colon cancer should be prioritized. The data from two key, likely landmark trials are eagerly awaited over the next few years, and these studies will likely change the standard of care for adjuvant chemotherapy in stage II and stage III colon cancer. The COBRA trial is currently recruiting and it specifically addresses whether ctDNA can serve as a predictive biomarker in adjuvant chemotherapy for patients with low-risk stage II colon cancer (Table 2). In addition, the CIRCULATE-US study is also recruiting and it is addressing whether ctDNA can guide adjuvant chemotherapy decisions for stage III colon cancer, most notably the ability to successfully provide long-term surveillance with ctDNA alone for those patients with negative ctDNA, as well as whether escalating to triplet chemotherapy (FOLFOXIRI) would be beneficial for those patients with positive ctDNA (Table 2). Fortunately, the advent of new molecular technologies such as ctDNA has brought optimism on the horizon for patients affected by this challenging disease and, hopefully, the next few years will offer improved clinical outcomes and avoid unnecessary treatment-related toxicities due to a refined therapeutic approach in adjuvant chemotherapy.

## Figures and Tables

**Figure 1 cells-12-01068-f001:**
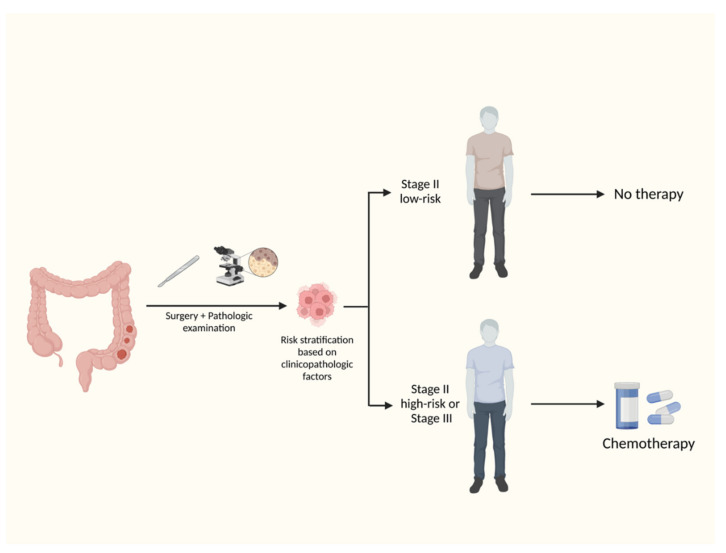
**Standard of care of adjuvant therapy of colon cancer.** The current selection of patients for adjuvant chemotherapy is based on the presence of prognostic clinicopathologic factors, which offer an inaccurate risk stratification in low-risk and high-risk stage II and stage III colon cancer. Created with Biorender.com.

**Figure 2 cells-12-01068-f002:**
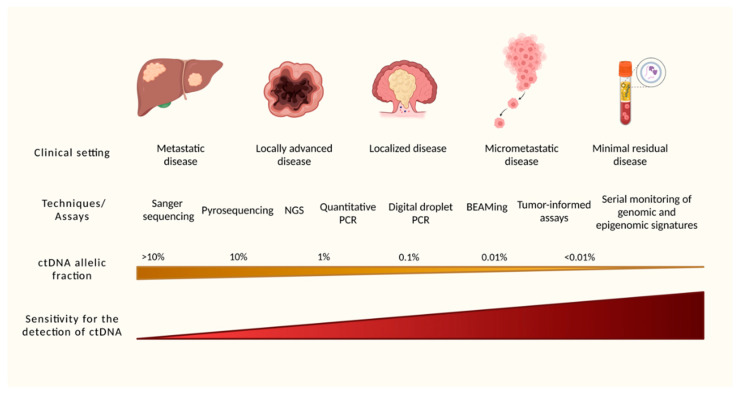
**Techniques and assays for detection of ctDNA.** The sensitivity of the molecular techniques has increased in recent years for the identification of small allelic fractions of ctDNA. The development of tumor-informed assays and association of epigenomic signatures have allowed detection of allelic fractions lower than 0.01% and with high specificity. Created with Biorender.com. *Abbreviations: NGS: next-generation sequencing, PCR: polymerase chain reaction*.

**Figure 3 cells-12-01068-f003:**
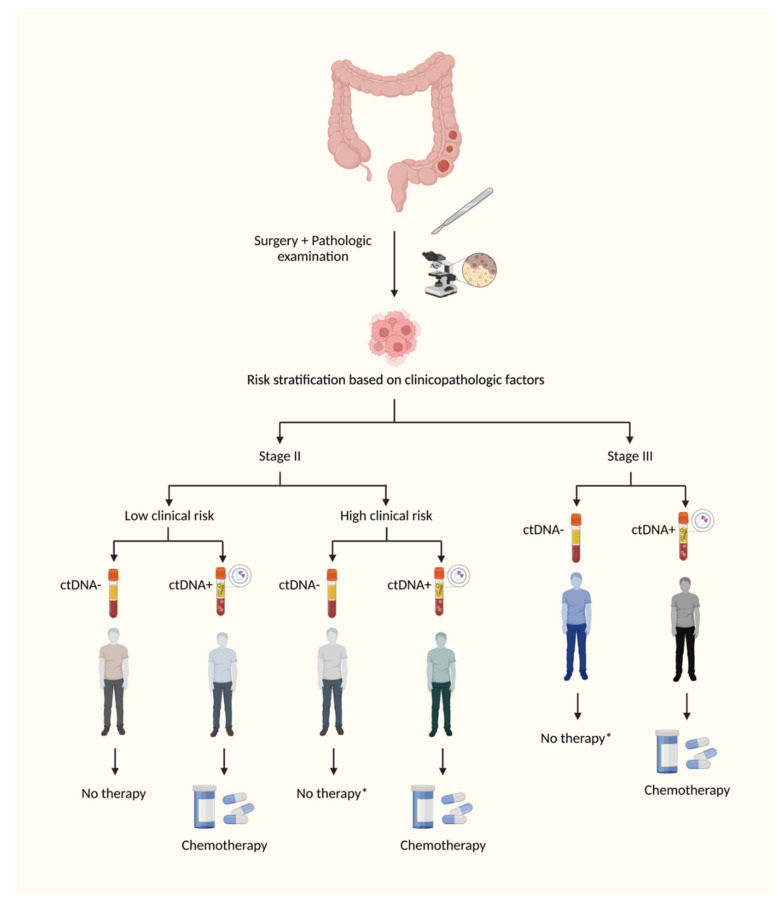
**Potential ctDNA-guided postoperative management.** This figure illustrates a potential therapeutic algorithm for the decision of adjuvant therapy in colon cancer after the advent of molecular techniques for the detection of minimal residual disease. Certain clinicopathologic factors, such as T4 tumors, have demonstrated significant prognostic value, even in ctDNA-negative patients. Based on the available clinical data, even with the improvement of molecular techniques, it is unlikely that the test will reach very high levels of sensitivity in the first postoperative weeks, when the decision to offer adjuvant chemotherapy has to be made. Hence, it is possible that the combination of molecular (ctDNA) and clinical data might be a reasonable strategy to offer a more personalized therapy for patients with localized colon cancer. * De-escalation of adjuvant chemotherapy in high clinical risk ctDNA-negative patients will only be possible after the development of techniques with very high sensitivity. Created with Biorender.com.

**Table 1 cells-12-01068-t001:** **Completed clinical studies evaluating the prognostic role of ctDNA as MRD in colorectal cancer**.

Author	N	Stage	Median Follow-up Time (m)	Assay	Risk of Recurrence in ctDNA+ Pts (*n*/N)	Risk of Recurrence in ctDNA- Pts (*n*/N)	HR RFS (95% CI)
Tie et al. [27]	230	II	27.0	Tumor-informed	11/14 (79%)	16/164 (10%)	18.0 (7.9–40.0)
Reinert et al. [24]	130	I–III	12.5	Tumor-informed	7/10 (70%)	10/84 (12%)	0.21 (0.06–0.69)
Tarazona et al. [28]	150	I–III	24.7	Tumor-informed	8/14 (57%)	9/55 (16%)	11.6 (3.6–36.8)
Parikh et al. [25]	103	I–IV	21.0	Plasma-only	15/17(88%)	12/49 (25%)	11.4 (NR)
Taieb et al. [26]	1017	III	79.2	Plasma-only	93/140 (66%)	673/877 (77%)	1.5 (1.1–2.1)
Tie et al. [29]	100	III	28.9	Tumor-informed	10/20 (50%)	14/76 (18%)	7.5 (3.5–16.1)
Kotaka et al. [30]	1050	I–IV	11.4	Tumor-informed	91/188 (48%)	52/852 (6%)	15.3 (8.6–27.2)
Marmorino et al. [31]	76	IV	77.0	Tumor-informed	33/39 (84%)	20/37 (54%)	1.8 (1.0–3.3)
Li et al. [32]	165	III	33.5	Tumor-informed	13/24 (54%)	26/127 (21%)	5.5 (2.4–12.3)
Lonardi et al. [33]	69	IV	NR	Tumor-informed	29/31 (94%)	19/38 (50%)	6.4 (3.0–14.0)
Tie et al. [34]	455	II	37.0	Tumor-informed	8/45 (18%)	15/246 (6%)	1.8 (0.7–4.2)

Abbreviations: MRD: minimal residual disease, m: months, HR: hazard ratio, RFS: recurrence-free survival, CI: confidence interval, NR: not reported.

**Table 2 cells-12-01068-t002:** **Ongoing clinical trials evaluating the clinical applicability of ctDNA as MRD in colorectal cancer**.

Population	N	Intervention in ctDNA+ Pts	Primary Objective	Identifier
Single-arm studies				
I–III	300	None	Rate of ctDNA+	NCT04726800
II–IV	15	CB-NK cells + cetuximab	ctDNA clearance rate	NCT05040568
II–IV	17	3-month aspirin + vitamin D + diet + physical activity	ctDNA clearance rate	NCT05036109
II–IV	15	Trifluridine/tipiracil	ctDNA clearance rate	NCT05343013
II–IV	15	CXCR1/2 inhibitor (SX-682) + nivolumab (STOPTRAFFIC-1)	ctDNA clearance rate	NCT04599140
II–IV with MRD after ACT	22	Trifluridine/tipiracil + irinotecan	Rate of ctDNA+ after CT	NCT04920032
IV after hepatectomy	120	FOLFOX or FOLFIRI ± bevacizumab	1-year RFS	NCT05062317
Randomized studies				
II–III with MRD after 3-month ACT	236	3-month FOLFOXIRI vs. 3-month FOLFOX/CAPOX	3-year DFS rate	NCT05534087
IIA (COBRA)	1408	6-month FOLFOX/CAPOX s. active surveillance	ctDNA clearance rate	NCT04068103
II–III with MRD after surgery (CIRCULATE-US)	1912	6-month FOLFOX/CAPOX vs. 6-month FOLFIRINOX	ctDNA clearance rate	NCT05174169

Abbreviations: CT: chemotherapy, ACT: adjuvant chemotherapy, MRD: minimal residual disease, RFS: recurrence-free survival, DFS: disease-free survival.

## Data Availability

Not applicable.

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
