# Peer review of "Minimal Residual Disease in Colorectal Cancer: Are We Finding the Needle in a Haystack?"

_cells, 2023, doi:10.3390/cells12071068_

Round 1

Reviewer 1 Report

The authors in the manuscript attempt a review on the importance of ctDNA in stratification and management of patients with colon cancer. They explain the existing pitfalls in the managing the patients that lead to over treatment or undertreatment during adjuvant therapies. They introduce the concept and use case of ctDNA that determining the MRD. They summarize methods and limitations that reliably detect ctDNA from different tissues/sources. Accurate detection of ctDNA is an evolving field, mostly dependent on the strategy and existing platform. This review makes an good attempt to sensitize cancer biologists and oncologists on importance of ctDNA in patient management.  

The manuscript is well written with good readability. However, there are issues that can be addressed to improve the overall value of the manuscript

Major comments

1.      Authors used colorectal cancer model in the current manuscript, however a brief overview on the progress of ctDNA in other cancers would be desirable

2.      Authors can also briefly discuss on using ctDNA in preclinical studies as a be a viable tool in drug discovery

Minor comments:

1.      Figure and Table labels missing in the legends

2.      Legends for Figure 2 and 3 need to be condensed

3.      The paragraph on MRD in rectal cancer could be added after the paragraph on MRD in metastatic colorectal cancer.

4.      Sentence in line 413 needs rephrasing

5.      Document should be checked for minor grammatical errors

Reviewer 2 Report

The authors provide a very interesting review on the state of the techniques to identify circulating allelic fractions of tumor DNA and the clinical trascendence of this finding in patients operated for Colorectal Cancer. The authors discuss the published scientific background and the publications that allow the interpretation of circulating fractions of tumor DNA as minimal residual disease (MRD) in patients operated for colorectal cancer. The possible utility of this assay in the adyuvent setting of this cancer patients is discused. Thepaper is well present.

Reviewer 3 Report

The authors have prepared a cogent and fairly comprehensive review on a topic of great and timely importance.  In general, I found the MS easy to read and understand.  However,  a few comments to consider in revision:

1) Reorganize Table 1: needs a heading, 'authors' put at rightmost column, 'assays' make first column.

2)  Does the last figure add anything new? To this reviewer it does not and can be removed.  

3) What is lacking are more detailed descriptions of the molecular techniques, who generates the data, time to generate and analyze the data,  cost/insurance and whether these types of analyses can only be done at major centers.

4) Similarly, little detail/explanation is provided regarding the target genes/sequences that are used and whether these are standardized across clinical trials.  Would this not be important?  Moreover, is this not evolving as more studies of genomic and epigenetic changes in CRC are published. 
